# Epigenetic Modulators as Therapeutic Agents in Cancer

**DOI:** 10.3390/ijms241914964

**Published:** 2023-10-06

**Authors:** Eshaan Patnaik, Chikezie Madu, Yi Lu

**Affiliations:** 1Department of Biology, Memphis University School, Memphis, TN 38119, USA; epatnaik6@gmail.com; 2Departments of Biological Sciences, University of Memphis, Memphis, TN 38152, USA; comadu@memphis.edu; 3Department of Pathology and Laboratory Medicine, University of Tennessee Health Science Center, Memphis, TN 38163, USA

**Keywords:** epigenetics, DNA methylation, cancer

## Abstract

Epigenetics play a crucial role in gene regulation and cellular processes. Most importantly, its dysregulation can contribute to the development of tumors. Epigenetic modifications, such as DNA methylation and histone acetylation, are reversible processes that can be utilized as targets for therapeutic intervention. DNA methylation inhibitors disrupt DNA methylation patterns by inhibiting DNA methyltransferases. Such inhibitors can restore normal gene expression patterns, and they can be effective against various forms of cancer. Histone deacetylase inhibitors increase histone acetylation levels, leading to altered gene expressions. Like DNA methylation inhibitors, histone methyltransferase inhibitors target molecules involved in histone methylation. Bromodomain and extra-terminal domain inhibitors target proteins involved in gene expression. They can be effective by inhibiting oncogene expression and inducing anti-proliferative effects seen in cancer. Understanding epigenetic modifications and utilizing epigenetic inhibitors will offer new possibilities for cancer research.

## 1. Introduction

Epigenetics refers to inheritable modifications in gene expression that occur without any changes to the DNA sequence itself. In normal cells, epigenetic modifications—histone modifications and DNA methylation—help regulate gene expression [1]. Gene silencing is crucial for various biological processes such as cellular identity, cellular maintenance, and environmental responses. Epigenetic processes that regulate the silencing of critical genes are involved in growth and division, contributing to genetic balance. However, aberrations in gene silencing can disrupt gene regulation, leading to the eventual formation of tumors or the inactivation of genes during normal aging [2]. Epigenetic silencing can impact the body the same way as genetic alterations, leading to cancer. Epigenetic modifications in a host cell can result in cellular growth, giving rise to the uncontrolled growth of a tumor. Unlike genetic changes, epigenetic changes are reversible, and by modulating these epigenetic marks, it is possible to restore normal gene expression patterns. For example, by targeting epigenetic marks, it is possible to inhibit tumor growth and reverse drug resistance, leading to enhanced treatment outcomes [3]. Advancements in epigenetic therapies can offer newfound potential in cancer treatment and other fields of medicine.

Epigenetic modulators activate or repress cancer-related epigenetic mechanisms [4]. Epigenetic modulators contain the following: DNA methyltransferase (DNMT) inhibitors; histone methyltransferases, demethylases, acetyltransferases, and deacetylases; and bromodomain and extra-terminal domain (BET) inhibitors [5]. These modulators serve to convert signals from cellular stresses to alter chromatin states at oncogenes or tumor suppressors and to promote epigenetic flexibility [4]. Modulators can bring about changes in chromatin landscapes, enabling the necessary epigenetic adaptability crucial for cancer genesis and progression.

## 2. DNA Methylation Inhibitors

DNA methylation is a major epigenetic modification that plays a significant role in gene regulation, cellular processes, and gene silencing. This process is catalyzed by DNA methyltransferases (DNMTs), which add a methyl group to the cytosine base located in Cytosine-phosphate-Guanine “CpG” dinucleotide sequences [6]. These CpG “islands” are prevalent in promoter regions of DNA, and aberrations have pointed towards cancer [7]. These DNTMs include DNMT1, DNMT 3A, and DNMT 3B. DNMT 3A and 3B are involved in de novo (new) methylation in which a new methylation pattern is created on unmethylated or hemi-methylated DNA [8]. On the other hand, DNMT 1, the “maintenance” methyltransferase, replicates the previous DNA methylation patterns during cell division [8]. DNMT disfunction can result in localized hypermethylation or global hypomethylation in certain regions along the DNA strand, resulting in genomic instability and varying epigenetic disorders [9]. It is worth noting that DNMT activity can be influenced by cellular age, with a decline in DNA methylation occurring as cells age [10]. This age-related decline in DNMT activity may contribute to age-associated changes in DNA methylation patterns.

DNMTs are widely overexpressed in various forms of cancer, and they are targeted agents for treating cancer [11]. DNMT inhibitors (DNMTis) are introduced into the DNA strand and covalently bind to DNMTs, limiting their enzymatic ability [12,13]. The disruption of methylation patterns can indirectly influence transcriptional factor (TF) activity by affecting the methylation status of regulatory proteins. These alterations can impact TF binding, leading to changes in cellular processes and behavior. Sp1, Sp3, E2F, and p53 transcription factors (TFs) regulate the transcriptional activity of DNMT promoters [14]. However, such DNMTis are associated with specificity and toxicity [15].

DNMTis are categorized into two families: nucleoside and non-nucleoside analogs. The most widely used nucleoside analogs, called azanucleosides, are 5′-azacytidine and 5-2′ deoxycytidine (decitabine) [16]. 

Instead of carrying an oxygen atom, azanucleosides substitute in a nitrogen atom, increasing the potential interactions between target molecules and azanucleosides. Overall, the mechanism of action for nucleoside analogs are similar, but there are subtle differences in their specific mechanisms. After they are incorporated into cells, azanucleosides, such as decitabine and azacytidine, included in Figure 1, undergo a series of metabolic conversions to become the active nucleotide form required for DNA methylation inhibition. Nucleotide transporters import azanucleosides into cells, followed by their activation by varying kinases [17]. The nucleoside is then subjected to ATP-dependent phosphorylation, creating a deoxyribonucleoside base. Azanucleosides are recognized as natural cytosine substitutes and substituted for a nucleotide base into the growing DNA strand [18]. The inclusion of DNMTis disrupts interactions between the DNA and DNMTs on the nitrogen atom of the azanucleoside, trapping DNMTs [19,20].

On the other hand, non-nucleoside analogues do not interact with genetic material. Figure 1 shows various non-nucleoside analogues. Some non-nucleoside analogues, such as procainamide, bind directly to the catalytic site of DNMT enzymes, competitively inhibiting their ability to interact with DNA [20]. These inhibitors can compete with S-adenosyl-L-methionine (SAM), the cofactor required for DNMT activity, for binding to the catalytic site. SAM, apart from its role as a cofactor, enhances the production of hydrogen sulfide through its activation of the enzyme cystathionine beta-synthase (CBS) [22]. Lower concentrations and shorter exposure to SAM can stimulate cell proliferation, while higher concentrations and longer exposure can inhibit proliferation [22]. It should be noted that the antitumor effects of SAM may involve a CBS-independent mechanism, suggesting that the specificity of SAM as an anticancer agent is limited. RG108, another type of non-nucleoside analogue, targets the active site of DNMTs. It exhibits specificity for DNMTs and reactivates silenced tumor suppressor genes while preserving the methylation status of centromeric repeats [23]. MG98 is an antisense oligonucleotide that targets DNMT1, initiating a methylation decrease [24]. Evidence shows that MG98 was effective in decreasing DNMT1 expression and restoring normal methylation patterns [25].

## 3. Histone Deacetylase Inhibitors

Histone modifications are post-translational modifications to histone proteins. DNA wraps around histone proteins, which serve as structural support for a chromosome. However, modifications such as acetylation, methylation, phosphorylation, ubiquitination, and sumoylation can affect gene activity or gene expression [26].

The histone acetylation of lysine is mediated by the action of two distinct families of enzymes, histone acetyltransferases (HATs) and histone deacetylases (HDACs), which play opposing roles in modulating histone acetylation levels [18]. The difference between HATs and HDACs lies in their enzymatic activities and the effects they have on histone proteins. HAT enzymes encourage lysine acetylation, whereas HDAC enzymes reverse the process [18]. By adding acetyl groups, HATs neutralize the positive charge of lysine and create a more relaxed chromatin structure [27]. Histone deacetylation involves the elimination of acetyl groups from lysine residues located in the NH2-terminal tails of core histones. This process results in a chromatin structure that is more compacted and leads to the repression of gene expression, countering the effects of HATs. This process is managed by HDACs which remove acetyl groups from lysine residues in histone [27] However, irregularities in the expression or activity of HDAC and HAT can result in abnormalities of their substrates, leading to tumor formation. HDACs are organized into four classes—I, II, III, and IV—based on their respective functions and sequences [18,28]. Class I HDACs are located in the nucleus and are involved in regulation gene expression by deacetylation histone proteins. Class 1 HDACs include HDAC1, HDAC2, HDAC3, and HDAC8. Class II HDACs include HDAC4, HDAC5, HDAC6, HDAC7, HDAC9, and HDAC10 [18,26,27,28,29]. They are required in cell cycle regulation, differentiation, and stress responses. Cell III HDACs, called sirtuins, include SIRT1-7. Sirtuins deacetylate both histone and non-histone proteins. Class IV HDACs share a similar structure to Class I and II HDACs [18,26,27,28,29].

In recent years, the discovery of histone deacetylase inhibitors (HDACis) has garnered significant attention in the field of cancer therapeutics. HDACis specifically target and inhibit the enzymatic activity of HDACs, leading to increased histone acetylation levels and altered gene expression patterns. Some HDACis have been shown to induce cell cycle arrest and cell death in cancer cells. They can trigger G1 phase arrest by upregulating p21 expression and dephosphorylating Rb [18]. In addition, some cell lines treated with HDACis exhibit G2 phase arrest. The selective toxicity of HDAC inhibitors is based on the integrity of the HDACi-sensitive G2 checkpoint, which is defective in certain tumor cells. As a result, these cells are more susceptible to cell death induced by HDAC inhibitors [28]. HDAC inhibitors can also synergize well with DNA-damaging agents and ionizing radiation-induced apoptosis [29].

In conjunction with other histone modifications, histone methyltransferase contributes to the complex regulation of gene activity and chromatin structure. Histone methyltransferases (HMTs), in contrast to HDACs, catalyze the addition of methyl groups to specific lysine or arginine residues onto histone proteins. HMTs are highly specialized, and they are grouped by specificity. The methylation of histone proteins involves the addition of one, two, or three methyl groups to such residues [30]. This change, while only affecting a small part of the primary structure of the protein, can have massive impacts on gene expression [31]. Different patterns of histone methylation can either promote or repress gene expression, potentially leading to cancer mutations. Methylation is performed by lysine methyltransferases (KMTs), which perform lysine methylation, and arginine methyltransferases (PRMTs), which perform methylation, with S-adenosylmethionine (SAM) serving as a co-factor or a methyl donor for both. Histone methylation can lead to either the activation or repression of gene expression.

## 4. Histone Methylation Inhibitors

The mechanism of lysine methylation varies based on the type of KMTs, referenced in Figure 2. Some KMTs can be characterized by the presence of the Su(var)3-9, Enhancer of Zeste, and Trithorax (SET) domain. 

The SET domain is a protein module responsible for the catalytic activity of HMTs. The arrangement of substrate binding sites in the SET domain, connected by a deep channel, is important for facilitating multiple rounds of lysine methylation without the dissociation of the protein substrate form the SET domain [32]. This allows SET domain methyltransferases to target multiple residues. The SET domain contains specific amino acid residues that interact with the target lysine residues, allowing the enzyme to bind to the histone substrate in a specific manner. SAM then binds to the SET domain, serving as the methyl donor group. Once the SET domain is bound to the substate and SAM, a series of enzymatic reactions occur, transferring the methyl group from SAM onto the e-amino group lysine residue [33]. After the methyl transfer, the methylated histone product is released from the enzyme, completing the enzymatic cycle [34]. The SET domain is released and is then available for subsequent rounds of methylation [31]. On the other hand, non-SET KMTs, as seen in Figure 2, employ other catalytic domains utilized by different lysine methyltransferases. The mechanism is nearly the same. Arginine methylation is similar to lysine methylation; however, there are key differences. Arginine methylation is catalyzed by PRMTs. Also, they do not contain specific domains like the SET domain; instead, they have distinct catalytic domains [35]. In addition, a methyl group from SAM is transferred to one of the nitrogens of the guanidine group in the arginine residue [36].

As a result of therapeutic cancer-treating development, HMTs are targeted. One version of therapeutics is targeting SET-domain proteins. As previously mentioned, the SET domain binds the cofactor SAM, and catalyzes the methyl transfer in a series of reactions. The structure of known SET domains varies, but their structures reveal that the SET domain can be targeted. Two regions of the protein are being targeted: the SAM-binding pocket and the substrate-binding pocket [37]. The domain itself is folded into small beta sheets arranged around a structure. This structure, resembling a knot or loop, is formed through the threading of the carboxyl terminus through a loop formed by a preceding segment, forming an active site next to the methyl donor [32]. Such regions in SET domain proteins are essential for the interaction and enzymatic functions of proteins. The inhibition of these regions in HMTs can be an effective strategy to regulate histone modulation and certain gene expression patterns.

Several examples of HMTs that are being targeted for therapeutics include EZH2, G9a, and DOT1L. EZH2, Enhancer of Zeste Homolog 2, is an oncogene that represses transcription [38]. It catalyzes the trimethylation of histone H3 at lysine 27. Evidence has shown that the overexpression of EZH2 can lead to tumor formations. Tazemetostat, an inhibitor of EZH2 activity, operates in a competitive mechanism against SAM [39]. Other examples include GSK126, UNC199, Pinometostat, and others. The noncovalent inhibitors of SAM have also been developed, targeting DOT1L. Instead of a SET domain, DOT1L has a SAM region of binding. DOT1L is targeted to actively transcribed genes and is responsible for the methylation of histone H3 at lysine 79. This process involves the binding of DOT1L to the phosphorylated D-terminal domain of RNA polymerase II [40].

N-methyltransferase (NNMT) is a cytosolic enzyme that regulates cellular energy homeostasis by regulating intracellular NAM and S-adenosyl-L-methionine SAM content [41]. SAM serves as a methyl donor for DNA and histone methylation reactions. By consuming SAM, NNMT can lead to the reduced availability of SAM for critical cellular processes. Increased NNMT activity in cancer cells lowers histone methylation and SAM levels [42]. The same is true for NAD+, reducing the amount of nicotinamide for NAD+ biosynthesis. NNMT has been reported to be overexpressed in several tumors, and its upregulation has been linked to cancer cell invasion and migration [43,44]. NNMT inhibitors are already available for targeting cancer and metabolic disorders. Cyclic peptides have potential for NNMT inhibition as well, and they could be the first allosteric inhibitors of NNMT [45]. GYZ-319 is one such potent NNMT inhibitor. However, the compound requires carboxylic acid and amine moieties to perform optimally and translate into strong cellular activity [45]. The discovery of a tricyclic core compound, known as JBSNT-000028, shows promise in NNMT inhibition. The compound did not exhibit activity against a broad panel of targets related to metabolism and safety, indicating a specificity for NNMT its potential as a therapeutic agent [46].

## 5. Bromodomain and Bet Inhibitors

A bromodomain (BRD) is a protein region that recognizes methylated residues in histone tails. A total of 61 BRDs have been identified in the human genome in 46 proteins that are categorized into eight subfamilies based on protein sequence similarities [47]. Every BRD molecule shares a conserved fold of four alpha helices that contribute to substrate specificity. The recognition of acetyl-lysine by BRDs involves a hydrophobic opening and a hydrogen bond with an asparagine residue present in many BRDs [48]. The conservation of specific amino acid residues, such as tyrosine and asparagine across different BRDs highlights their respective role in recognizing acetyl lysine [49]. Specifically, the acetylated lysine is recognized by the BRD in a pocket formed by two loops (ZA and ZB), which can vary in length and composition, influencing the specificity of different BRDs. Three amino acid residues inside these loops interact with the acetyl group through hydrogen bonding [50]. By binding to acetylated histones, BRDs can serve as a binding site for other proteins that contain functional domains capable of recruiting various protein complexes to specific genomic regions. Bromodomain proteins are found in a wide range of proteins, including transcription factors, chromatin remodelers, and other regulatory proteins. As a result, they are involved in key cellular processes, including the development of cancers. Understanding the function and regulation of bromodomains is critical for the development of therapeutics.

Bromodomain protein 4 (BRD4) is a member of the bromodomain and extra-terminal domain (BET) family of proteins. Essential to maintaining chromatin structure, the protein binds to acetylated histone 3 and histone 4 tails [51]. BRD4 contains two N-terminal bromodomains and an ET domain [52]. BRD4 is known for its role in oncogene expression regulation and super-enhancers (SEs). SEs are associated with genes that control cell state in previously identified cells. They are also associated with RNA polymerase II, and there is evidence of high levels of RNA surrounding SE regions, indicating that SEs are related to gene activation [53]. 

However, super enhancers, such as the one depicted in Figure 3, are linked to disease. Diseases linked to single nucleotide polymorphisms often manifest within the super-enhancers of cells relevant to the specific disease [54]. BRD4 promotes the assembly and maintenance of the SE structure [55]. BRD4 has a role in transcription repression, indicated by the LSD1/NuRD complex. The LSD1/NuRD complex and BRD4 colocalize at super enhancers and repress certain genes [56]. BRD4 has also been linked to the recurrent t(15;19) chromosomal translocation. The BRD4 breakpoints in each cancer provide evidence of a fusion oncogene mechanism involving BDR4 [57]. BDR4 also plays an important role in cell cycle regulation. BRD4 is implicated in transcriptional memory and the maintenance of cellular identity as it remains attached to chromosomes during mitosis [58].

Bromodomain protein 3 (BRD3) is a part of the BET protein family. BRD3 contains two bromodomains that enable BRD3 to interact with chromatin, and the gene localizes to 9q34 [59]. BRD3 has been found to play a role in nucleosome remodeling [60]. In addition, BRD3 can promote the transcription of oncogenic genes such as AKT, PI3K, and MYC [61]. BRD3 has also been linked to the NUT midline carcinoma. The cancer results from a fusion of BRD4 or BRD3 on chromosome 15q14. The fusion event occurs due to a chromosomal translocation, where a portion of the NUT gene fuses with the coding sequence of either BRD4 or BRD3 [62]. The misfunction of BRD3 is also linked to various neurological disorders, inflammatory diseases, and cardiovascular diseases [63,64]. BRD3 and BRD2 are closely related members of the BET family. Using its two bromodomain regions, BRD2 binds to the lysine-12 residue of histone H4 [60]. BRD2 is an integral part of the Mediator complex, a molecular bridge between transcription factors and RNA polymerase II [65,66].

BET inhibitors (BETis) a class of drugs that specifically target BET proteins such as BRD2, BRD3, and BRD4. The first known inhibitor of BET proteins was the ligand JQ1, a thieno-triazolo-1,4-diazepine, which was tested in NUT midline carcinoma [48]. JQ1 binds competitively to the acetyl-lysine area, disrupting the BRD4 oncoprotein. Results show that the displacement of BRD4 led to anti-proliferative effects and squamous differentiation in cells [49]. JQ1 has been found to be effective in solid tumors such as medulloblastoma, hepatocellular carcinoma, colon cancer, pancreatic cancer, prostate cancer, lung cancer, and breast cancer [67]. The drug can also induce apoptosis, decrease cell proliferation, and enhance radiosensitivity [68]. However, JQ1 has a short half-life and low oral bioavailability [69,70]. Since then, BETis have been created to target BET members that influence oncogenes such as MYC, JUNB, and CCND1 [71]. Other examples of BETis include PFI-1, a drug that similarly has anti-proliferative effects on cells. G1 cell cycle arrest, the downregulation of MYC expression, and apoptosis are notable effects of the PFI-1 inhibitor [72]. PFI-1 has a high selectivity and potency for BRDs. Many are linked to sensitive cell lines [66]. The inhibitor downregulates the Aurora B kinase (AURKB) expression, which is involved in mitosis and a therapeutic target in cancers [66,73]. AURKB could be a potential target in future combination therapy strategies. Aurora B kinase is primarily localized to the centromeres and inner centromeres of chromosomes during mitosis. It functions together with other proteins in the chromosomal passenger complex, which is used for chromosome lining, separating, and cytokinesis [74].

## 6. Current Challenges and Discoveries

Epigenetic modifications are linked to a variety of cellular processes, and targeting specific genes while avoiding off-target effects remains an obstacle. Ensuring that epigenetic therapy only affects the desired genes without impacting essential functions is key for patient safety. Making sure that inventions only target the intended genomic loci is important as traditional epigenetic drugs cause large scale changes in gene expression [67,68,75]. Chemoresistance poses a challenge in cancer research, limiting the effectiveness of traditional cancer treatments. Recently, in combination therapy, epigenetic drugs with different mechanisms of action have emerged as methods to address chemoresistance and increase treatment efficacy [63]. For example, combining the flexibility of the three-component HDAS inhibitor pharmacophore with other targets allows for many possible epigenetic inhibitors [76]. Delivery methods also pose an issue for current epigenetic drugs. Some drugs included in Table 1—such as romidepsin and vorinostat—suffer from toxicity and inefficient delivery to tumor sites, affecting their efficacy [77,78]. Another issue is durability and long-term stability. Because epigenetic modifications can be reversible, the long-term stability of therapeutic effects has been a concern. Issues such as abnormal pathway activation and high toxicity are the main factors behind durability [79]. The drugs approved by the FDA are mostly HDAC inhibitors [73]. The research of epigenetic drugs as therapeutic agents is still ongoing. There are risks behind such drugs. Some epigenetic drugs can lead to cytotoxicity in the body [75]. Cytotoxicity is one of the main reasons epigenetic drugs need to undergo thorough testing. Also, the continued use of drugs can lead to drug resistance, causing tumor recurrence and poor prognosis [80].

The recent emergence of the CRISPR/Cas 9 system has become prevalent in designing therapies. It is used in genomic silencing and transcriptional activation and repression [81]. The tool utilizes a molecule called guide RNA to direct Cas9, a protein, to a specific DNA sequence [79,82]. Then, the gRNA-Cas9 complex meets the corresponding DNA sequence. Thereafter, Cas9 introduces a double-strand break in the DNA molecule [83]. The cell’s natural DNA repair mechanisms become involved. By controlling the repair mechanism of the cell, researchers can introduce genetic changes into the genome. CRISPR tools have enabled scientists to engineer and correct specific epigenetic aberrations with a high degree of precision. Researchers can activate or suppress gene expression in a controlled fashion, which could also lead to more personalized treatments. However, while CRISPR technologies have the potential for targeted manipulation of epigenetic shifts, epigenome editing is still in its early stages. Genomic editing has made significant progress, whereas epigenomic editing has not. Precision is one of the main issues. For example, a study’s focus of acetylation of histone H3 at lysine 27 through the p300 enzyme shows the potential error in the system, raising concerns about potential off-target effects and antibody specificity [84].

Advances in nanotechnology also offer opportunities to develop more effective delivery systems. The use of nanoparticles as a potential vector for epigenetic drugs is promising. Nanoparticles, due to their unique properties, hold promise for the delivery of therapeutic agents. Stimuli—responsive nanoparticles, which release drugs under certain conditions like temperature and pH—could be useful [85]. Currently however, nanocarriers have a limitation of non-specific biodistribution, leading to low selectivity. They are only suitable for certain types of cancer [81]. Moreover, the immune system might not recognize nanoparticles as friendly, thereby enhancing the immune response [86].

Epigenetic biomarkers are being considered as a form of treatment for cancer. They are extremely important for understanding tumor development and subtyping. As a result of the environment influencing gene expression, epigenetic biomarkers can be useful in determining the susceptibility of individuals to cancer and their response to certain treatments [87]. Markers offer insight into the natural progression of diseases. Their stability in fluids such as plasma, serum, and urine and different tissue preparations like fresh, frozen, dried blood spots, and FFPE [88]. Ultimately, harnessing these epigenetic insights could revolutionize personalized cancer care.

Cancer risk assessment is perhaps the most important factor in cancer prevention. The prior detection and eventual mitigation of cancer risk is key by using less damaging treatment options than chemotherapy. The epigenome is both tissue-specific and condition-specific [88]. This leads to issues because the risk markers must be present in bodily fluids, such as blood or saliva for detection [89]. By analyzing specific epigenetic modifications associated with increased cancer risk, healthcare providers can offer targeted screening and surveillance associated with high-risk individuals. This will also be useful for low-risk individuals as they can be offered care without any substantial concern for progression of the disease.

## 7. Conclusions

Epigenetic modifications are crucial factors that are influenced by the environment and lifestyle, with potential implications for disease development. Epigenetic modifications play a significant role in various diseases, including cancer, neurological disorders, and autoimmune conditions. Luckily, the reversible nature of epigenetic modifications has opened new possibilities for developing new treatment strategies and therapeutics. By targeting specific enzymes involved in DNA methylation or histone modifications, it is possible to alter the epigenetic landscape and promote beneficial changes in gene expression. This approach holds promise for developing targeted therapies that can selectively modulate gene activity, potentially leading to more effective and personalized treatments for a wide range of diseases. Unlike genetic changes, which are permanent alterations in the DNA sequence, epigenetic changes can be modified or deleted, leading to the restoration of normal gene expression patterns and opportunities for therapeutic interventions. Currently, epigenetic drugs are used worldwide to treat cancer, particularly myelodysplastic syndromes and leukemias. However, they are limited in the treatment of solid tumors, and the development of resistance also poses significant challenges. In addition, broad-scale gene alterations caused by traditional epigenetic medicines can lead to the re-expression of improperly silenced genes, including oncogenes and premetastatic genes. There is an urgent need for novel approaches in epigenetic therapy. Finding ways to introduce more locus-specific alterations to the epigenome and identifying more therapeutic profiles are key challenges to overcome. By addressing such challenges, treatment outcomes will improve.

In order to address specific needs, researchers are currently understanding and testing new epigenetic drugs with improved features such as enhanced safety profiles and longer half-lives. Scientists are looking into synergizing epigenetic drugs with chemotherapy to sensitize cancer cells and increase responsiveness to chemotherapy. This approach has shown the significant suppression of tumor growth, even in solid tumors. Additionally, combining different epigenetic drugs can overcome chemoresistance in drugs and improve drug efficacy. The development of multitargeting drugs is another idea being explored. The discovery of metabolic enzymes that can alter the epigenome has also opened a new avenue for drug development. The use of CRISPR/dCAS9 technology for targeted epigenetic therapy holds great potential. The future of research is most likely to focus on natural product-based non-nucleoside inhibitors. The development of new inhibitors and exploration of alternative scaffolds will also be expected to enhance the field of epigenetics.

## Figures and Tables

**Figure 1 ijms-24-14964-f001:**
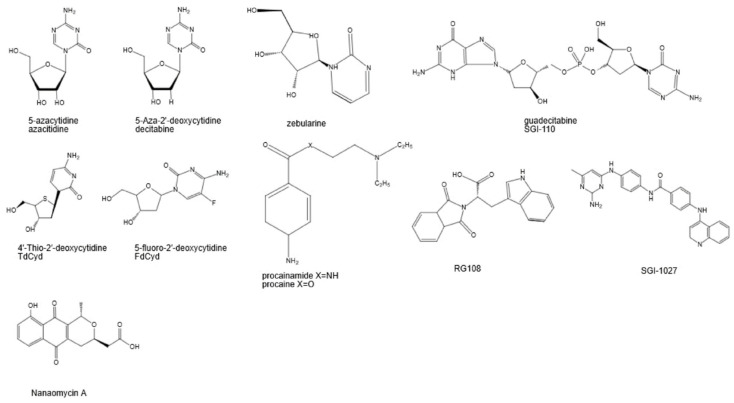
Various structure of non-nucleoside and nucleoside analogs adapted from Ref. [21].

**Figure 2 ijms-24-14964-f002:**
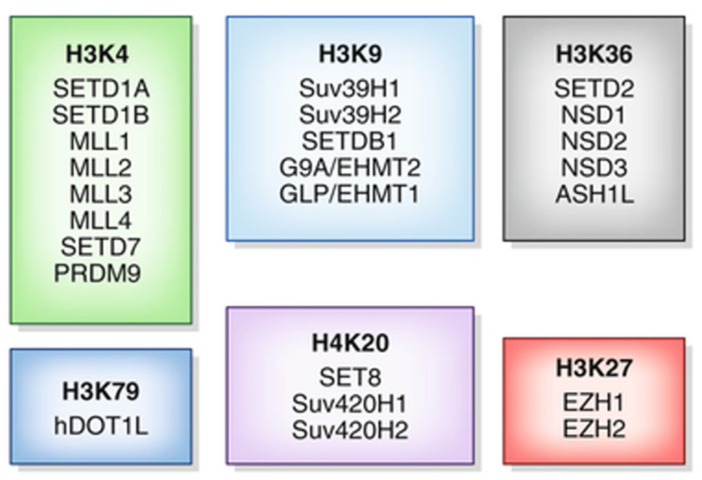
KMTs separated by specificity adapted from Ref. [30].

**Figure 3 ijms-24-14964-f003:**
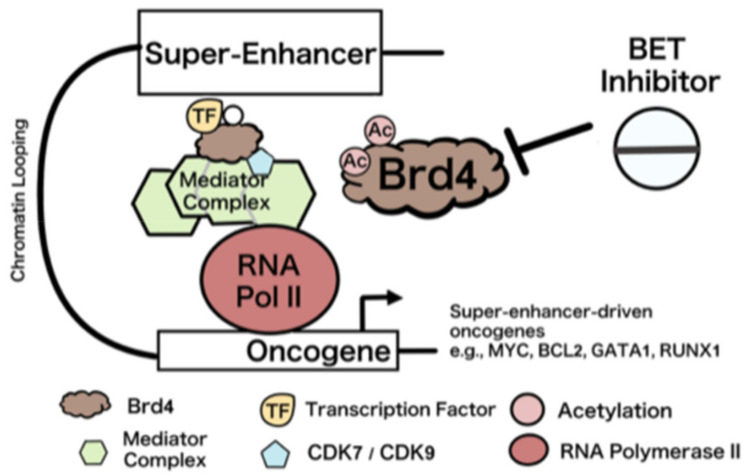
BRD4 interacting with super-enhancer in a cell adapted from Ref. [57].

**Table 1 ijms-24-14964-t001:** An overview of epigenetic drugs

Drug	Formula	Target	Status	Uses/Concerns/Other
Azacytidine	C_8_H_12_N_4_O_5_	DNMT inhibitor	FDA-approved	Myelodysplastic syndromes
Decitabine	C_8_H_12_N_4_O_4_	DNMT inhibitor	FDA-approved	Myelodysplastic syndromes
Pseudoisocytidine	C_9_H_13_N_3_O_5_	DNMT inhibitor	Discontinued	Hepatotoxicity concerns
DHAC	C_8_H_14_N_4_O_5_	DNMT inhibitor	Discontinued	Cardiotoxicity concerns
Guadecitabine	C_18_H_23_N_9_O_10_P	DNMT inhibitor	Discontinued	Lack of Phase 3 efficacy
Vorinostat	C_14_H_20_N_2_O_3_	HDAC inhibitor	FDA-approved	Cutaneous T-cell lymphoma
Romidepsin	C_24_H_36_N_4_O_6_S_2_	HDAC inhibitor	FDA-approved	Cutaneous T-cell lymphoma
Butyric acid	C_4_H_8_O_2_	HDAC inhibitor	Awaiting	Exploring inhibition in models
Belinostat	C_15_H_14_N_2_O_4_S	HDAC inhibitor	FDA-approved	Peripheral T-cell lymphoma
Panobinostat	C_21_H_23_N_3_O_2_	HDAC inhibitor	FDA approval withdrawn	Peripheral T-cell lymphoma
Entinostat	C_21_H_20_N_4_O_3_	HDAC inhibitor	Paused	Lack of Phase 3 efficacy
Tucidinostat	C_22_H_19_FN_4_O_2_	HDAC inhibitor	CFDA-approved	Peripheral T-cell lymphoma
Pinometostat	C_30_H_42_N_8_O_3_	HMT inhibitor	Discontinued	Lack of efficacy
Tazemetostat	C_34_H_44_N_4_O_4_	HMT inhibitor	FDA-approved	Relapsed/refractory follicular lymphoma
Pemrametostat	C_24_H_32_N_6_O_3_	HMT inhibitor	Paused	Clinical trial paused

## Data Availability

Not applicable.

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
