# Peer review of "Epigenetic Modulators as Therapeutic Agents in Cancer"

_ijms, 2023, doi:10.3390/ijms241914964_

Round 1

Reviewer 1 Report

The manuscript “Epigenetic Modulators as Therapeutic Agents in Cancer” is a review regarding epigenetic modifications in cancer and related possible markers for targeted therapy. The manuscript is generally well written and can be of interest for the readers. However, authors cover only partially the topic and several flaws are present, thus the manuscript cannot be accepted for publication in this form.

Authors are strongly encouraged to improve the manuscript accordingly.

1.       why the title of some paragraph ends with the double dots.

2.       Figure 2 is somehow chaotic and difficult to be interpretated. A colored and more linear figure will help the readers.

1.       The main concern is that this manuscript completely ignores a master regulator of intracellular NAD and SAM content, namely the nicotinamide N-methyltransferase (NNMT) which can influence a number of enzymes involved in epigenetics, such as histone deacetylases sirtuins (PMID: 36829935). This enzyme has been reported to be overexpressed in a variety of solid tumors, where it contributes to the tumorigenicity and aggressiveness (PMID: 34827592; PMID: 34439880). Since NNMT can affect NAD homeostasis, NAD-dependent enzymes and concentration of SAM, it has a great impact on epigenetics, as demonstrated by Ulanovskaya et al. in an elegant study (PMID: 23455543). Moreover, it seems that NNMT can impact the metabolism of the cell (PMID: 37014628; PMID: 36750850)

Notably, a number of NNMT inhibitors are already available and seems to be a promising strategy for targeted therapy in cancer and for metabolic disorders (PMID: 34572571; PMID: 34704059; PMID: 34424711; PMID: 36104373).  All these considerations should be included since they would let the manuscript cover totally the topic of epigenetic modulators in cancer.

3.       A table summarizing the inhibitors would greatly help the readability of the manuscript.

4.       The figure 3 contains something (deriving from Powerpoint software perhaps?!), in the upper part that should be deleted.

English is fine.

Author Response

1.  why the title of some paragraph ends with the double dots.

Response: The issue of the incorrectly formatted titles has been corrected.

2. Figure 2 is somehow chaotic and difficult to be interpretated. A colored and more linear figure will help the readers.

Response: Figure 2 has been further simplified and colorized.

3. The main concern is that this manuscript completely ignores a master regulator of intracellular NAD and SAM content, namely the nicotinamide N-methyltransferase (NNMT) which can influence a number of enzymes involved in epigenetics, such as histone deacetylases sirtuins (PMID: 36829935). This enzyme has been reported to be overexpressed in a variety of solid tumors, where it contributes to the tumorigenicity and aggressiveness (PMID: 34827592; PMID: 34439880). Since NNMT can affect NAD homeostasis, NAD-dependent enzymes and concentration of SAM, it has a great impact on epigenetics, as demonstrated by Ulanovskaya et al. in an elegant study (PMID: 23455543). Moreover, it seems that NNMT can impact the metabolism of the cell (PMID: 37014628; PMD: 36750850) Notably, a number of NNMT inhibitors are already available and seems to be a promising strategy for targeted therapy in cancer and for metabolic disorders (PMID: 34572571; PMID: 34704059; PMID: 34424711; PMID: 36104373).  All these considerations should be included since they would let the manuscript cover totally the topic of epigenetic modulators in cancer.

Response: A comprehensive section has been included about NNMT in the passage prior to “BROMODOMAIN AND BET INHIBITORS.” The passage includes an overview of NNMT and its relevance to epigenetics and therapies. Sources 45-50 address the issue.

4. A table summarizing the inhibitors would greatly help the readability of the manuscript.

Response: A new figure (Table 1) has been created. This figure describes the FDA approved epigenetic drugs that are currently available. This table includes the target, formula, and clinical uses of each drug.

5.The figure 3 contains something (deriving from Powerpoint software perhaps?!), in the upper part that should be deleted.

Response:  Figure 3 has been corrected.

Reviewer 2 Report

The manuscript summarizes the most relevant epigenetic inhibitors in oncology. The review is interesting and it is well written and comprehensive. However I have several concerns:

1.- Some tiitles are ended by double dots

2.- A table and a description of FDA approved epigenetic inhibitors would be informative

3.-A section about side effects and limitations of these inhibitors should be included

4.- A table with the different clinical trials with epigenetic inhibitors should be added

English is fine

Author Response

1. Some titles are ended by double dots

Response: The titles have been corrected. They no longer end with double dots.

2. A table and a description of FDA approved epigenetic inhibitors would be informative

Response:  A table has been included at the end of the manuscript. Figure 4 details the FDA approved drugs and includes a review of each drug, its clinical uses, chemical formula, and target.

3. A section about side effects and limitations of these inhibitors should be include

Response:  A section about the side effects and the limitations of the inhibitors has been included. Sources 30, 83-88 correct this issue.

4. A table with the different clinical trials with epigenetic inhibitors should be added

Response:  Sentences of various clinical trials have been added in the manuscript detailing the various clinical trials of epigenetic inhibitors.

Round 2

Reviewer 1 Report

The authors improved the manuscript by addressing all concerns so the manuscript can be published.

Minor editing of English language required

Author Response

RE: Manuscript ID: ijms-2615438 Final Revision

The responses to the reviewer’s comments are listed below, which contain corrections/modifications we made according to this reviewer’s (as well as all other reviewers’) comments:

-Figure 1 has been moved to after reference 18. Fig. 1 is cited in the text (on line 5 of the 1st paragraph on page 5, and on line 5 from the bottom, page 5).

-Previous Figure 3 has been changed to the current figure 2. It is now located before reference 34. Fig. 2 is cited in the text (2nd line on page 8 and 1st line on page 9).

-Previous Figure 2 has been changed to the current figure 3. It is now located after 57. Fig. 3 is cited in the text i(n the beginning sentence of the last paragraph on page 11).

-Table 1 has been modified. It now includes a column about status and uses/concerns/other. These two columns detail FDA approval or trial status. Some epigenetic drugs have been labeled as Approved, discontinued, or paused. Uses/concerns/other details the (if the drug has been approved) clinical uses, the concerns for why the drug was discontinued, and special cases if the drug is awaiting or paused for clinical trials. It has now been moved near the end of the paper, after reference 88.

Sincerely,

Eshaan Patnaik, Chikezie Madu, Yi Lu